# EgoLM: Multi-Modal Language Model of Egocentric Motions

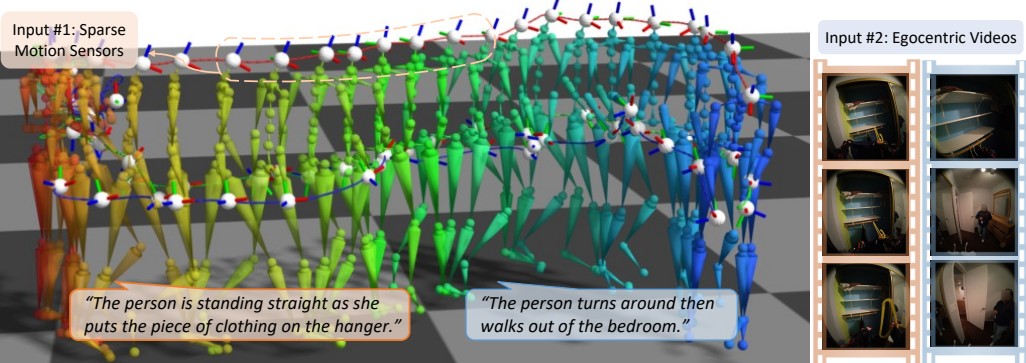

Figure 1: We propose **EgoLM**, a multi-modal language model that unifies egocentric motion tracking and understanding from wearable sensor data, *i.e.*, sparse motion sensors and egocentric videos.

## Abstract

As wearable devices become more prevalent, understanding the user's motion is crucial for improving contextual AI systems. We introduce **EgoLM**, a versatile framework designed for egocentric motion understanding using multi-modal data. EgoLM integrates the rich contextual information from egocentric videos and motion sensors afforded by wearable devices. It also combines dense supervision signals from motion and language, leveraging the vast knowledge encoded in pre-trained large language models (LLMs). EgoLM models the joint distribution of egocentric motions and natural language using LLMs, conditioned on observations from egocentric videos and motion sensors. It unifies a range of motion understanding tasks, including motion narration from video or motion data, as well as motion generation from text or sparse sensor data. Unique to wearable devices, it also enables a novel task to generate text descriptions from sparse sensors. Through extensive experiments, we validate the effectiveness of EgoLM in addressing the challenges of under-constrained egocentric motion learning, and demonstrate its capability as a generalist model through a variety of applications.

## 1 Introduction

Smart wearable devices, such as Ray-Ban Meta (Meta, 2024) and Spectacles (Snap, 2024), offer new opportunities for developing personal AI assistants by capturing the world from the user's perspective. They provide real-time egocentric observations about the user's environment, interactions, and actions. On the other hand, large language models (LLMs) (Brown et al., 2020; Touvron et al., 2023) encode such context through text in their latent space, which can be leveraged for common-sense reasoning and human understanding. The fusion of egocentric perception and common-sense reasoning presents a unique and exciting opportunity for advancing contextual AI research, among which, egocentric motion understanding is an essential task (Plizzari et al., 2023).

However, a key challenge in utilizing egocentric perception is the **lack of direct observations of the wearer**. Two types of observations are available from wearable devices, *i.e.*, *1) egocentric videos* and *2) sparse motion sensors*. Egocentric videos, captured by cameras mounted on smart glasses, provide rich contextual information of the wearer's environment and interactions. But the wearer's body is rarely visible in the video, due to constrained camera mounting position and angle. Sparse

motion sensors provide low-level kinematic motion of a few important body parts, *i.e.*, head motions from glasses and wrists movements from smart watches. However, they are insufficient to inform the full body pose, especially lacking information of the lower body.

Our insight is that these **two types of indirect observations are complementary to each other**. Egocentric videos can provide strong clues of the environment, and help disambiguate the lower body motion. For example, a laptop placed on an office table is a strong indication that the wearer is sitting rather than squatting. Sparse motion sensors, on the other hand, offer precise tracking of important body parts, such as hand movements, which can help in scenarios where no body part is visible in the video. For example, sparse motion sensors can differentiate between jumping jacks and simple jumps, where egocentric video may appear identical.

Another key challenge in egocentric human understanding is **aligning motion and language representations**, so that we can leverage the vast contextual knowledge embedded in LLMs to describe motion. While motion signals are continuous, low-level kinematic representations, natural language consists of unstructured and discrete tokens. To bridge this gap, we **treat motion as a form of language**. By tokenizing motion and repurposing a pre-trained LLM to model the joint distribution of motion and language, we facilitate an effective alignment between these two distinct representations.

With the above insights, we introduce **EgoLM**, a versatile framework for egocentric motion understanding that leverages rich sensor observations and strong contextual understanding from LLMs. As shown in Fig. 1, EgoLM takes sparse motion sensor data and egocentric videos as inputs, and is capable of generating motion and natural language as output. The framework unifies a range of motion understanding tasks, at both the *kinematic* and *semantic* levels. At the kinematic level, EgoLM can perform motion tracking from three-point (Jiang et al., 2022) or one-point (Li et al., 2023) sensor data, incorporating egocentric videos for disambiguation. At the semantic level, EgoLM can generate motion narration from various combinations of input modalities. More importantly, we highlight a novel task of motion narration from three-points and egocentric videos, unique to AR use cases.

Compared with recent VLMs (Liu et al., 2023b;a), our approach tackles a more complex and challenging problem involving **more modalities and tasks with greater disparities**. In particular, both our input modalities and output tasks encode information at varying levels of granularity. To tackle it, we employ **multi-modal multi-task joint training** through instruction tuning. Multiple input modalities are aligned to LLM latent space with rich contextual information, and interleaved between text instructions. Multi-task training exploits connections between tasks and benefits each other. For instance, three-points motion tracking bridges the gap between sparse motion sensors and natural languages, improving the performance of motion narration from three-points and videos.

To validate the proposed framework, we perform extensive experiments on a large-scale motion dataset, Nymeria (Ma et al., 2024). Compared with previous dedicated motion tracking and understanding models, we show better performance in both tasks, under different combinations of input modalities, proving EgoLM as a generalist model. Our contributions are summarized below.

**1) We introduce a egocentric motion generalist model EgoLM**, which integrates a variety of motion understanding tasks at both kinematic and semantic levels. By leveraging large language models (LLMs), we aim to enhance egocentric perception, thereby contributing to the advancement of contextual AI research. **2) We address the challenge of under-constrained egocentric motion learning** by combining two complementary modalities, *i.e.*, sparse motion sensors and egocentric videos. This new paradigm enables two unique applications for AR use cases: *motion tracking and narration from sparse motion sensors and egocentric videos*. **3) We employ multi-modal multi-task joint training to bridge substantial gaps between modalities and tasks.** Extensive experiments validate the effectiveness of this training strategy.

## 2 RELATED WORK

**Motion Regression.** Many efforts are devoted to regress 2D or 3D keypoints from human images or videos (Toshev & Szegedy, 2014; Martinez et al., 2017; Pavllo et al., 2019; Loper et al., 2023).Wearable motion sensors are also used for motion capture (Ponton et al., 2023; Mollyn et al., 2023; Milef et al., 2023; Yi et al., 2023; Jiang et al., 2023). Recent advancements in VR/AR have developed a new setup for motion tracking (Du et al., 2023; Jiang et al., 2022; Castillo et al., 2023; Li et al.,

Table 1: **Comparison with Related Works.** EgoLM uses novel techniques to effectively unify a wide range of multi-modal motion understanding tasks. "Vid.": egocentric videos. "Mot.": motions.

| Method | Motion Tokenizer | LM Type | Pre-Training | Instruction Tuning | Modalities | | | |
| --- | --- | --- | --- | --- | --- | --- | --- | --- |
| | | | | | 3pts | 1pt | Mot. | Vid. |
| LLaVA | N/A | Decoder-Only | N/A | Image Understanding | | | | ✓ |
| MotionGPT | Vanilla VQ-VAE | Encoder-Decoder | Motion-Text Pairs | Motion-Text Translation | | | ✓ | |
| EgoLM (Ours) | Product Quantization Motion VQ-VAE | Decoder-Only | Motion Only | 3pts/1pt/Vid. Motion Tracking 3pts/Mot./Vid. Motion Narration | ✓ | ✓ | ✓ | ✓ |

2023), *i.e.*, three-points and one-point body tracking. In this work, we target motion tracking from sparse sensors and rich semantics in egocentric videos to disambiguate under-constrained cases.

**Motion Generation.** There have been many efforts in generating motions from various conditions, *i.e.*, action labels (Petrovich et al., 2021; Guo et al., 2020), natural languages (Zhang et al., 2024; Tevet et al., 2022; Punnakkal et al., 2021; Guo et al., 2022a; Zhang et al., 2023b; Guo et al., 2022b). Recently, researchers use powerful LLMs to model the joint motion-language distribution for text-to-motion generation (Zhang et al., 2023c; Zhou et al., 2023). In EgoLM, we also adopt the similar idea. But in comparison with MotionGPT (Jiang et al., 2024), as listed in Tab. 1, EgoLM improve the motion tokenizer, employ the more scalable decoder-only LM, does not rely on paired data for pre-training and support more egocentric motion tasks and modalities.

**Motion Understanding.** There have been many setups in motion understanding. From the input side, human videos, either from third-person view (Soomro et al., 2012; Kuehne et al., 2011; Tran et al., 2015; Wang et al., 2016; Yan et al., 2018) or first-person view (Damen et al., 2021; 2022; 2018), are used for this task. From the output side, action recognition has been a classic task (Soomro et al., 2012; Damen et al., 2018). More recently, with the development of LLMs, some researches also propose to use natural languages as output (Jia et al., 2022; Xu et al., 2024; Grauman et al., 2022; Xue et al., 2023; Chen et al., 2023). In EgoLM, we highlight a new setup of motion narration from sparse motion sensors and egocentric videos, that is unique to AR use cases.

**Language Models.** LLMs have been a huge success in recent years with the large-scale pre-training (Radford et al., 2019; Brown et al., 2020) and alignment (cha, 2022; Achiam et al., 2023). To exploit the powerful text generation ability, image (Liu et al., 2023b;a) or video understanding (Zhang et al., 2023a) are defined as conditional text generation. LLaVA (Liu et al., 2023b) proposes to encode images with pre-trained vision encoders (Radford et al., 2021) and perform instruction tuning with LLMs (Touvron et al., 2023). EgoLM adopts the similar idea to tackle the challenge of large modality and task gaps. As shown in Tab. 1, compared with LLaVA, EgoLM handles a more complex egocentric setup, with more modalities and tasks with larger disparities.

## 3 METHOD

The overview of EgoLM is demonstrated in Fig. 2. There are three key steps in EgoLM training. In the first step, we train a motion VQ-VAE as the motion tokenizer (Sec. 3.2). The second step is motion pre-training for motion distribution learning (Sec. 3.3). The last step is multi-modal multi-task joint training to guide the model to perform various egocentric motion tasks (Sec. 3.4).

### 3.1 PRELIMINARIES

**Language Model.** Language models (LMs) model the distribution of natural languages. Recent breakthroughs in LMs suggest the effectiveness of the transformer-based architecture (Vaswani et al., 2017). A normal LM consists of three parts. The first is a codebook that stores the embeddings for each text token. The second part is the transformer backbone that takes text embeddings as inputs. Output features are mapped to probabilities of the next tokens by the third part, LM head.

**Motion Representation.** Human motions are represented as sequences of poses, global translations and rotations defined on the root joint. Each frame of pose is represented by joint angles, defined on a kinematic tree. For better learning of motion dynamics, we also include joint angle velocity in the representation. To avoid the normalization of global translation, we use the translation velocity $V_t^r \in \mathbb{R}^3$ for each frame, which can be integrated back to global translations. To ease the regression difficulty of rotation angles, we use 6D rotation representations (Hempel et al., 2022) for the root rotation $R_t^r \in \mathbb{R}^6$, root rotation velocity $R_t^{rv} \in \mathbb{R}^6$, joint angles $R_t^j \in \mathbb{R}^{22 \times 6}$, and joint angle

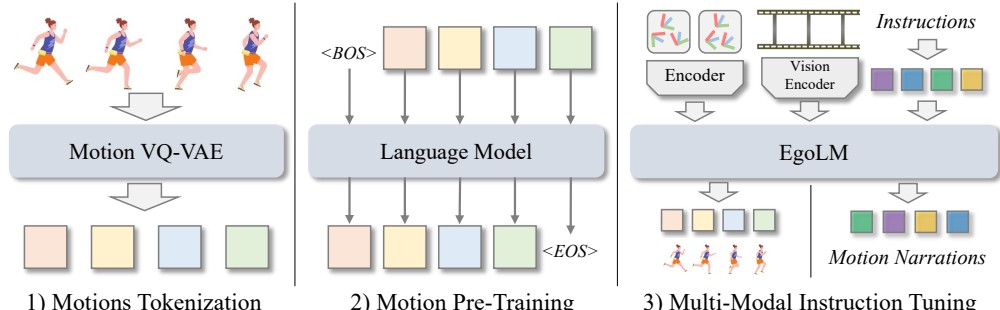

1) Motions Tokenization      2) Motion Pre-Training      3) Multi-Modal Instruction Tuning

Figure 2: **Overview of EgoLM.** Three steps are designed for the training of EgoLM, *i.e.*, motion tokenizer training, motion pre-training and multi-modal instruction tuning.

velocity $R_t^{jv} \in \mathbb{R}^{22 \times 6}$. Formally, we represent human motions with $T$ frames as $M = \{P_t\}_{t=1}^T$, where $P_t = [V_t^r; R_t^r; R_t^{rv}; R_t^j; R_t^{jv}] \in \mathbb{R}^{279}$. Forward kinematics (FK) together with integration of root velocity can be used to recover the joint positions $J = \text{FK}(M) \in \mathbb{R}^{23 \times 3}$.

### 3.2 MOTION TOKENIZER

To treat the motion as a form of a language and train with LMs, we first need a motion tokenizer, which can be realized by VQ-VAE (Oord et al., 2017). As shown in Fig. 3 a), the motion VQ-VAE consists of a fully convolutional encoder $\mathcal{E}$ and decoder $\mathcal{D}$. The fully convolutional design enables processing motions with arbitrary lengths. The encoder embeds raw motion representation to latent features $f^m = \mathcal{E}(M)$, where $f^m \in \mathbb{R}^{T/r \times c}$, $M \in \mathbb{R}^{T \times 279}$. $r$ is the down-sample rate.

Then, codebooks are learned to quantize the motion latent features. To ensure high-fidelity motion tokenization, we use three quantization techniques, which are 1) exponential moving average (EMA), 2) codebook reset (Dhariwal et al., 2020), 3) product quantization (Jegou et al., 2010; Lucas et al., 2022). The first two techniques increase the usage rate of codebooks. Product quantization increases the codebook expressiveness by decomposing the latent space into a Cartesian product of sub-spaces with lower dimensions. Specifically, the latent feature $f^m$ is split equally into $N$ trucks $\{f_n^m\}_{n=1}^N$, which are quantized separately by $N$ codebooks $\{Z_n\}_{n=1}^N$. Each codebook with $K$ entries is defined as $Z_n = \{z_i\}_{i=1}^K$, where $z_i \in \mathbb{R}^{c/N}$. The quantization process for feature $f_{tn}^m$ at frame $t$ and trunk $n$ is formulated as

$$i_{tn} = Q(f_{tn}^m) = \arg\min_{z_i \in Z_n} \|f_{tn}^m - z_i\|_2. \tag{1}$$

The resulting indices $i_{tn}$ are flattened and used as motion token sequences $W = \{[(i_n)_{n=1}^N]_t\}_{t=1}^{T/r}$, which has the length of $L_W = N \times (T/r)$. After quantization, we obtain the corresponding codebook entry for the motion latent feature $\hat{f}^m = \{\hat{f}_t^m\}_{t=1}^{T/r} = \{z_{i_t}\}_{t=1}^{T/r}$. It is input into the decoder $\mathcal{D}$ to decode raw motion representation $\hat{M} = \mathcal{D}(\hat{f}^m)$.

For the training of VQ-VAE, two types of training losses are used. The first is the commitment loss $\mathcal{L}_c = \|f^m - \hat{f}^m\|_2$ for the codebook learning. The second is motion reconstruction loss $\mathcal{L}_r$, which consists of raw representation loss $\mathcal{L}_m$, joint position loss $\mathcal{L}_j$, rotation velocity loss $\mathcal{L}_v$, which are defined as

$$\begin{aligned}
\mathcal{L}_r = \lambda_m \mathcal{L}_m + \lambda_j \mathcal{L}_j + \lambda_v \mathcal{L}_v &= \lambda_m \|M - \hat{M}\|_1 + \lambda_j \|\text{FK}(M) - \text{FK}(\hat{M})\|_1 \\
&+ \lambda_v \|R_{1:T-1}^{rv} - (R_{1:T-1}^r)^{-1} R_{2:T}^r\|_1 + \lambda_v \|R_{1:T-1}^{jv} - (R_{1:T-1}^j)^{-1} R_{2:T}^j\|_1.
\end{aligned} \tag{2}$$

We define the smoothed L1 loss as $\| \cdot \|_1$. In summary, the training loss of the motion VQ-VAE is $\mathcal{L}_{vq} = \lambda_c \mathcal{L}_c + \lambda_r \mathcal{L}_r$, where $\lambda_*$ are manually adjusted weights.

### 3.3 MOTION PRE-TRAINING

EgoLM aims to empower egocentric motion learning with strong prior in pre-trained LMs. However, the pre-trained LM only models the distribution of natural languages. Therefore, to facilitate motion generation, we perform motion pre-training with LM to learn motion distributions.

Figure 3: **Details of a) Motion Tokenizer (VQ-VAE) and b) Motion Pre-Training.** Product quantization provides high-fidelity motion tokenization. It is used for motion pre-training with a decoder-only LM, where codebook and LM head extension are in need.

Before pre-training LM with motion tokens, two modifications are in need, as shown in Fig. 3. Firstly, since the pre-trained LM only contains embeddings for text tokens, we expand the LM codebook in accordance with the size of motion codebook. Secondly, the output shape of the LM head is also expanded accordingly. Using the motion tokenizer described above, motion representations $M$ can be encoded and flattened to a sequence of motion tokens $W = \{w_i\}_{i=1}^{L_W}$. They are fed into the LM to learn the motion distribution by conducting the next-token prediction (Radford et al., 2019). Specifically, we maximize the log-likelihood of the next-token probability given the previous token inputs and network parameter $\Theta$. The loss function $\mathcal{L}_{pre}$ is formulated as

$$\mathcal{L}_{pre} = -\sum_{i=2}^{L_W} \mathbb{P}(w_i|w_1...w_{i-1};\Theta). \tag{3}$$

As the by-product of this stage training, we obtain an auto-regressive motion generator. Given a leading motion sequence as the prompt, it can sample an arbitrary length of human motions that continues the given motion. More importantly, the LM learns human motion distributions and has the ability of sampling plausible human motions, which lays a solid foundation for the next stage.

## 3.4 MULTI-MODAL MULTI-TASK JOINT TRAINING

As previously discussed, EgoLM addresses a more complex and challenging problem, involving multiple modalities and tasks with significant disparities. On the modality side, in addition to motion and natural languages, we need to integrate data from sparse motion sensors and egocentric videos, which capture information at varying levels of granularity. Furthermore, EgoLM approaches egocentric motion understanding tasks from both kinematic and semantic perspectives. To tackle the challenge, we propose to employ multi-modal multi-task joint training to bridge the gaps between modalities and uncover the inherent connections between tasks.

Recent research on multi-modal LLMs has demonstrated that instruction tuning (Liu et al., 2023b; Achiam et al., 2023; cha, 2022; Zheng et al., 2023) effectively aligns different modalities and integrates multiple tasks. In our approach, various modalities are encoded differently. For motions and natural languages, both serve as inputs and outputs; thus, they are tokenized for auto-regressive modeling. Sparse motion sensors and egocentric videos are used exclusively as inputs. It is more efficient to encode these into continuous features that align with the LM latent space. Different tasks are differentiated by text instructions. Specifically, the instruction template typically includes: 1) text instructions specifying the tasks to perform; 2) inputs relevant to the task; and 3) expected outputs. Below, we provide two instruction examples for motion tracking and narration.

| | |
|---|---|
| **Task:** *Motion Tracking* 
 **Instruction:** *Perform motion tracking based on the given three-points and CLIP embeddings.* 
 **Input:** *Input CLIP embeddings:* `<CLIP_Placeholder>`. *Input three-points feature:* `<TP_Placeholder>` 
 **Output:** `<Motion_Placeholder>` | **Task:** *Motion Narration* 
 **Instruction:** *Describe the human motion based on the given three-points and CLIP embeddings.* 
 **Input:** *Input CLIP embeddings:* `<CLIP_Placeholder>`. *Input three-points feature:* `<TP_Placeholder>` 
 **Output:** `<Narration_Placeholder>` |

The encoded three-points 6-DoF poses would replace `<TP_Placeholder>`. The placeholder for egocentric video features is `<CLIP_Placeholder>`. Motions are encoded to tokens and filled in `<Motion_Placeholder>`. `<Narration_Placeholder>` is the placeholder for corresponding motion narration. A detailed illustration of how we organize different modalities of data is shown

*" Perform ... based on the given ... Input CLIP embeddings: <CLIP_Placeholder>. Input three-points: <TP_Placeholder>"*

Figure 4: **Details of Multi-Modal Instruction Tuning.** Different modalities are encoded separately. Their features are concatenated in the order of the instruction template and input into the transformer layers of the language model.

in Fig. 4. Texts are tokenized and embedded to feature vectors through LM embedding. Egocentric videos are sampled to sequences of frames and encoded by CLIP image encoder (Radford et al., 2021), which are further projected by linear layers to the LM feature space. Similarly, sparse motion sensor data, *e.g.*, sequences of three-points 6-DoF poses, is encoded by a fully convolutional encoder. Lastly, all the encoded features are concatenated in an interleaved way and input into the transformer layers of the LM.

With instruction templates established for each task, we can facilitate joint training across the following tasks: a) motion tracking with three-points and egocentric videos, b) motion narration using three-points and egocentric videos, c) text-to-motion generation, and d) motion-to-text generation. During training, these four tasks are randomly sampled with equal probability. The loss function utilized is the next-token prediction loss, as defined in Eq. 3.

During inference, natural language is sampled in the same manner as LMs for motion narration tasks. For motion tracking, our auto-regressive modeling offers the advantage of online inference. At each new time step, the incoming data is concatenated with historical data and fed into EgoLM. A single feed-forward inference is then performed to obtain the motion token for the current time step. For further details, please refer to the appendix.

## 4 EXPERIMENTS

### 4.1 EXPERIMENTAL SETUP

**Dataset.** We use the Nymeria dataset (Ma et al., 2024) to train and validate our method. The dataset includes: **a)** full-body motions captured by the Xsens Mocap system (Roetenberg et al., 2009), **b)** egocentric videos recorded with Aria glasses (Somasundaram et al., 2023), and **c)** motion narrations by human annotators. Three-point 6-DoF poses are derived from ground truth joints for comparison with prior work. The motion tracking training set comprises $147.89$ hours of data, with a test set of $41.93$ hours. For motion understanding, the training set includes $16,673$ segments (totaling $15.77$ hours), while the test set contains $7,468$ segments (totaling $6.76$ hours).

**Training Details.** Motion VQ-VAE utilizes two codebooks, each containing $8,192$ entries with a code dimension of $64$. The down-sample rate is set to $r = 4$. For motion tracking, all experiments use a window size of $60$ frames (equivalent to $1$ second), with random rotation augmentation applied to the motions. We employ GPT-2 Medium (Radford et al., 2019) as the language backbone.

**Evaluation Protocols.** For motion tracking, we calculate joint position errors (for full, upper and lower body), joint angle errors (for full body and root joint). For motion narration, the outputs are natural languages. Therefore, we adopt NLP metrics, including BERT (Zhang et al., 2019), BLEU (Papineni et al., 2002), and ROUGE (Lin, 2004) scores. For more details about the evaluation protocols, please kindly refer to the appendix.

### 4.2 MOTION TRACKING

**Quantitative Results.** We present th quantitative results of motion tracking in Tab. 2. All methods are evaluated using batch inference, where every 60 frames are processed independently. We assess various input combinations from three modalities, *i.e.*, three-points 6-DoF poses ("3pts"), one-point 6-DoF poses ("1pt") and egocentric videos ("Vid"). In the 3pts-only and 1pt-only settings, EgoLM demonstrates performance comparable to task-specific algorithms, highlighting the effectiveness of LMs for precise motion tracking. Additionally, we incorporate egocentric videos to provide contex-

Table 2: **Quantitative Results of Motion Tracking.** EgoLM performs comparably with task-specific algorithms. Incorporating video input can outperform methods without. "Full", "Upper", "Lower" are joint position errors in $mm$. "J.A.", "Root" are joint angle errors for full body and root joint in degree. [†]We directly replace three-points with one-point to train AvatarPoser.

| Method | Input Modality | | | Full | Upper | Lower | J.A. | Root |
|---|---|---|---|---|---|---|---|---|
| | 3pts | 1pt | Video | | | | | |
| AvatarPoser (Jiang et al., 2022) | ✓ | | | 85.89 | 52.78 | 165.18 | **12.41** | 14.78 |
| Bodiffusion (Castillo et al., 2023) | ✓ | | | 79.80 | 52.79 | 152.68 | 12.74 | **13.09** |
| Ours | ✓ | | | 83.88 | 54.06 | 148.37 | 13.31 | 14.13 |
| Ours | ✓ | | ✓ | **73.38** | **49.67** | **124.58** | 12.48 | 13.23 |
| AvatarPoser[†] (Jiang et al., 2022) | | ✓ | | 129.23 | 94.19 | 192.34 | 16.55 | 21.60 |
| EgoEgo (Li et al., 2023) | | ✓ | | 132.16 | 100.02 | 190.32 | 18.90 | 21.80 |
| Ours | | ✓ | | 127.45 | 97.87 | 174.92 | 16.97 | 20.57 |
| Ours | | ✓ | ✓ | **106.95** | **83.73** | **141.26** | **14.67** | **19.04** |

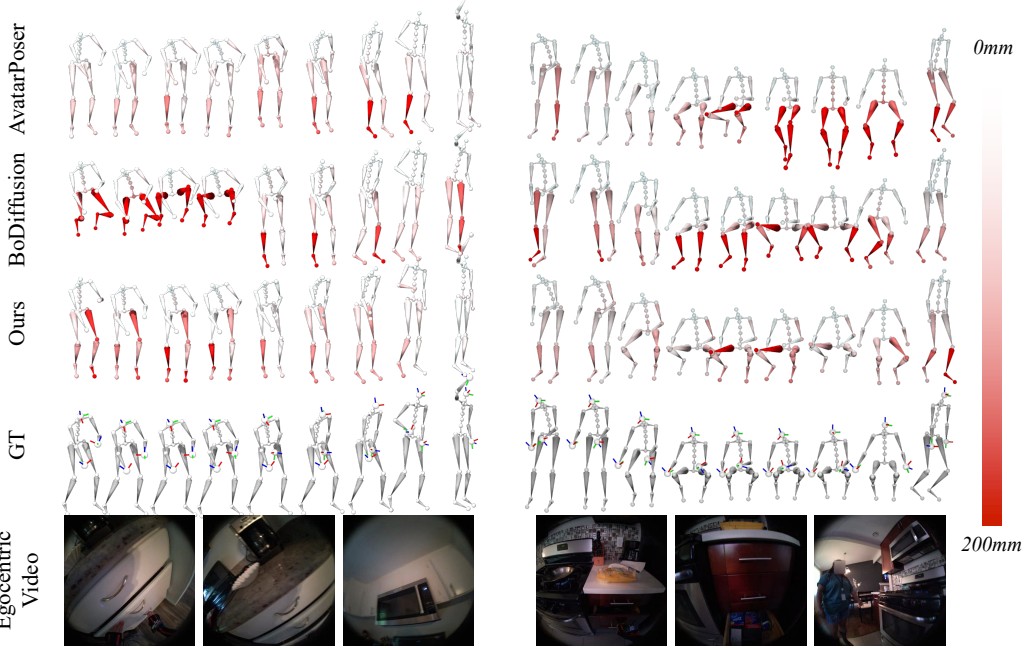

Figure 5: **Qualitative Results of Three-Points Motion Tracking.** Skeletons are color-coded by the joint position errors. Baseline methods use 3pts as inputs. Ours uses 3pts and videos as inputs.

tual information for motion tracking. For three-points tracking, this additional modality results in a 10 mm improvement in full-body joint error. For one-point tracking, the inclusion of egocentric videos leads to a 20 mm reduction in joint error, underscoring their effectiveness in disambiguating the ill-posed problem.

**Qualitative Results.** The results and comparisons for three-point motion tracking are presented in Fig. 5. Due to the inherent ambiguity, AvatarPoser incorrectly generates standing poses for squatting sequences (right example). BoDiffusion, while capable of producing correct results in some instances (*e.g.*, the squatting example), also faces ambiguity issues, as demonstrated in the bending-down sequence (left example). These examples highlight the importance of contextual consideration in motion tracking for effective disambiguation. Our full model reliably performs three-point body tracking in these challenging scenarios.

The results for one-point motion tracking are presented in Fig. 6. This task is particularly challenging for upper body tracking. As in left example, the upper body motions generated by EgoEgo significantly diverge from the ground truth. In the right example, EgoEgo mistakenly produces sitting poses for standing frames and vice versa, illustrating the ambiguity issue. In contrast, egocentric videos not only help to resolve this ambiguity but also provide clues about hand positions. In the

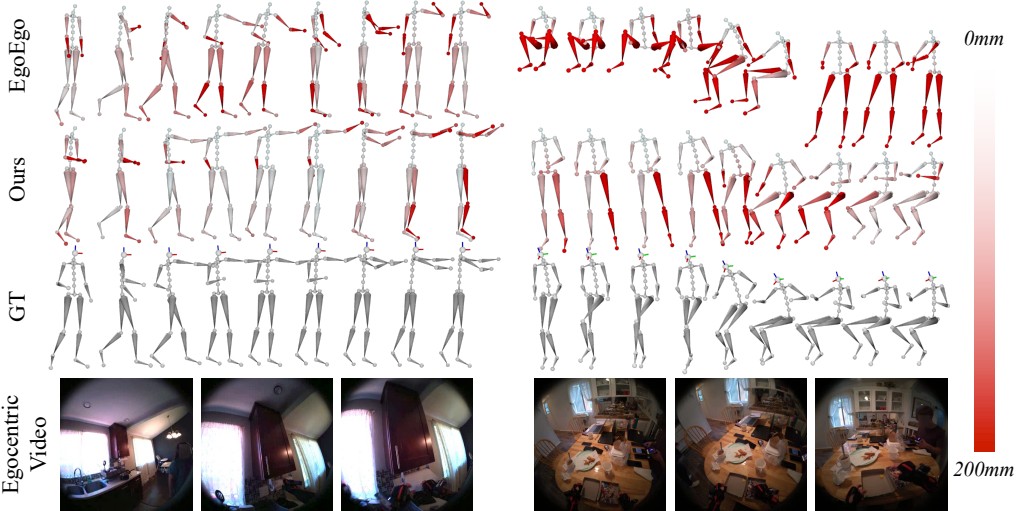

Figure 6: **Qualitative Results of One-Point Motion Tracking.** Skeletons are color-coded by joint position errors. EgoEgo only uses one-point as inputs. Ours includes egocentric videos as inputs.

left example, when hands are visible in the frames, our model leverages vision clues to capture this information and generate accurate arm movements. More visual results are provided in appendix.

## 4.3 MOTION NARRATION

**Quantitative Results.** We report the quantitative results of motion narration in Tab. 3. This task involves three input modalities, *i.e.*, three-points ("3pts"), motions, and egocentric videos ("Vid"). with various combinations evaluated. We first compare EgoLM with two existing motion narration methods that utilize motion as their sole input, *i.e.*, TM2T (Guo et al., 2022b) and MotionGPT (Jiang et al., 2024). TM2T trains language generation from scratch and consequently exhibits poor performance. MotionGPT leverages a pre-trained T5 model (Raffel et al., 2020). EgoLM(M2T&T2M) outperforms these methods, benefiting from the scalability of its decoder-only architecture. When we combine egocentric videos with motion inputs (MV2T&T2M), we achieve the best overall performance, as this combination offers comprehensive information for motion narration.

Using motion as input requires precise motion tracking, which is not always feasible, prompting us to explore sensor inputs instead. We tested two variants: three-points-only (TP2T) and egocentric videos only (V2T). The TP2T variant demonstrated a noticeable drop in performance compared to the motion-only version, as three-points provide limited information about body motion. Conversely, the V2T variant outperformed the motion-only version because egocentric videos capture relevant environmental context for our motion narrations. This underscores the significance of egocentric videos in understanding motion.

We then evaluate our highlighted setup of combining three-points and egocentric videos for motion narration. There are three approaches to achieve this. The first involves integrating two existing setups: 1) three-points motion tracking and 2) motion-to-text generation (TPV2M + MV2T). This variant shows a slight performance drop compared to MV2T due to error accumulation and requires a time-consuming two-pass inference. The second approach directly trains a three-points plus egocentric videos to text generation model (TPV2T) using our proposed multi-modal instruction tuning. While this outperforms using only egocentric videos or motions, it still lags behind the MV2T variant due to missing lower body information. To address this, we propose joint training of four tasks to establish connections between three-point poses and motion narrations, achieving optimal performance in a single forward pass for this new task.

**Qualitative Results.** We show four examples of motion narration in Fig. 7. TM2T and MotionGPT use full body motions as inputs, while our model incorporates three-points and egocentric videos. TM2T's language generation is trained from scratch, leading to frequent errors and nonsensical outputs. MotionGPT generates reasonable descriptions; for instance, in the lower left example, it correctly identifies the motion as "removing a piece of clothing from the hanger". However, our target motion narration is closely tied to environmental context, which TM2T and MotionGPT struggle with due to the absence of visual signals. In contrast, although EgoLM does not directly use

Table 3: **Quantitative Results of Motion Narration.** Different input modality combinations are tested. All metrics are higher the better.

| Method | Input Modality | | | Bert↑ | Bleu@1↑ | Bleu@4↑ | RougeL↑ |
|---|---|---|---|---|---|---|---|
| | 3pts | Motion | Video | | | | |
| TM2T (Guo et al., 2022b) | | ✓ | | 11.08 | 40.11 | 8.99 | 30.70 |
| MotionGPT (Jiang et al., 2024) | | ✓ | | 14.09 | 42.22 | 10.31 | 32.33 |
| Ours (M2T&T2M) | | ✓ | | 15.90 | 42.68 | 11.06 | 33.71 |
| Ours (MV2T&T2M) | | ✓ | ✓ | **20.32** | **45.33** | **12.80** | **35.31** |
| Ours (TP2T) | ✓ | | | 11.94 | 41.70 | 9.85 | 31.47 |
| Ours (V2T) | | | ✓ | 16.62 | 43.03 | 11.34 | 33.13 |
| Ours (TPV2M + MV2T) | ✓ | | ✓ | **19.97** | **45.41** | **12.81** | **35.04** |
| Ours (TPV2T) | ✓ | | ✓ | 18.38 | 44.55 | 12.12 | 33.80 |
| Ours (Joint Training) | ✓ | | ✓ | **19.40** | **45.45** | **12.74** | **34.82** |

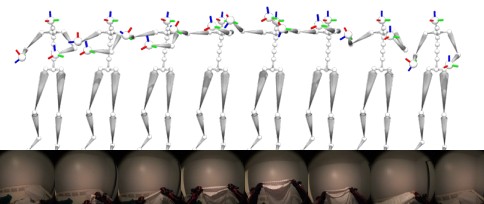

*TM2T: The person is sitting at the table as he lays her body on the sofa then leans backwards while talking and looking at her colleague. The person is resting both of her arms on her lap, lifts and bends both of her arms as she sits down on the sofa. The person is sitting on the sofa with both legs bent and slightly spread apart.*

*MotionGPT: The person is standing still in front of the sofa while holding a piece of clothing. The human's left arm is bent and raised upward with his left hand holding a piece of clothing. The human is standing with both legs apart and both feet resting on the floor.*

*Ours: The human is standing in the bedroom to fold the piece of clothing. The human is folding the piece of clothing with his left and right hand. The person is resting his left and right foot on the floor.*

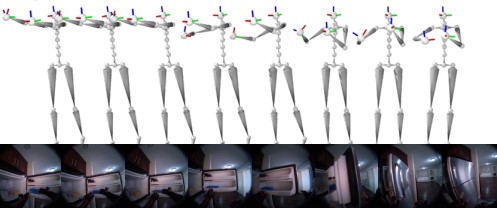

*TM2T: the person stands up straight as she holds the pillow and place them on the table. the person then arrange the pillow in the middle of the room with her right hand and places it on the table, while her left arm is slightly bent in front as she holds and arrange the pillow in the direction of the table.*

*MotionGPT: The person stands in front of the cabinet to remove the clothes from the hanger. the human raises both of his arms to remove a piece of clothing from the hanger. the human stands with both feet fixed on the floor.*

*Ours: The person is standing by the refrigerator while putting the pack of food inside the freezer. The human puts the pack of food inside the freezer with her right hand as her left hand holds the refrigerator door. The human is standing with both feed fixed on the floor.*

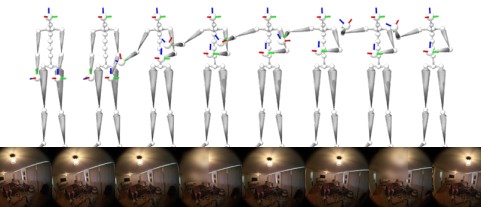

*TM2T: The person is standing still in front of the cabinet while making a hanger. The person bends and raises her left hand then lays the hanger on her side of her chest then spreads both arms on her side below her chest. The person stands with both legs stretched upright and both feet fixed on the floor.*

*MotionGPT: The person is standing straight at the living room ... The human has both arms naturally hanging at her sides then she bends, extends and raises her right arm and throws the object on the living room with her right hand. ... The human has both feet fixed on the floor with both legs stretched upright then she slightly bends and spreads both of her legs widely apart.*

*Ours: The person is standing still in the living room while talking to her peer. The human lifts both of his arms and then moves both hands in circular motion as she gesticulates. The human rests both of his feet on the ground.*

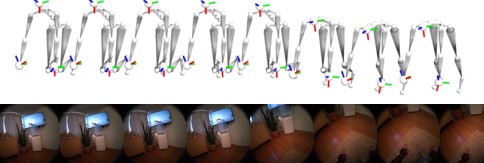

*TM2T: The person lowers her thigh as she lays down on the floor while kneeling on the floor. The person extends both her arms as she moves her right elbow on the floor to support her body.*

*MotionGPT: The human bends down while kneeling in the living area. The person extends both of her arms on the floor to support her body. The human extends both of her legs on the ground.*

*Ours: The person bends down as she planks on the floor. The human extends both of her arms on the floor to support her body. The person extends both of her legs while tiptoeing both of her feet.*

Figure 7: **Qualitative Results of Motion Narration.** We use green to highlight correct parts and red for mistakes.

motions as inputs, it jointly models the distributions of different modalities, enabling it to generate accurate narrations based on varying scenarios. Please kindly refer to appendix for more qualitative results results.

### 4.4 ABLATION STUDY

**Window Size of Motion Tracking.** As shown in Tab. 4, increasing the window size for three-points motion tracking from 60 to 120 frames results in an improvement of 4.2 mm in joint position errors. This enhancement is expected, as a larger window size provides more context, aiding disambiguation. When egocentric videos are included, further improvements are observed. Notably, using 60 frames with egocentric video outperforms using 120 frames alone, suggesting that the context provided by egocentric videos is more effective than simply increasing the window size.

Table 4: Ablation Study on Window Size for Motion Tracking.

| Win | Vid | Full | Upper | Lower | J.A. |
|---|---|---|---|---|---|
| 60 | | 83.88 | 54.06 | 148.37 | 13.31 |
| 120 | | 79.61 | 52.66 | 138.87 | 13.01 |
| 60 | ✓ | 73.38 | 49.67 | 124.58 | **12.48** |
| 120 | ✓ | **72.76** | **49.20** | **123.09** | 12.52 |

Table 5: Ablation Study on Reconstruction Results of Motion VQ-VAE. [$mm$]

| PQ | CB | Dim | MPJPE | PA-MPJPE | ACCEL |
|---|---|---|---|---|---|
| 1 | 2048 | 512 | 51.60 | 37.55 | 1.09 |
| 2 | 2048 | 512 | 39.63 | 29.77 | 0.71 |
| 2 | 16384 | 256 | 39.13 | 29.78 | 1.08 |
| 2 | 16384 | 64 | **34.49** | **26.83** | **0.67** |

Table 6: Ablation on the LM size. Medium: 345M; Large: 1.5B

| GPT-2 Size | Medium | Large |
|---|---|---|
| Bert↑ | 18.38 | **19.56** |
| Bleu@1↑ | **44.55** | 44.48 |
| Bleu@4↑ | 12.12 | **12.49** |
| RougeL↑ | 33.80 | **35.21** |

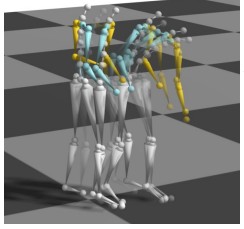 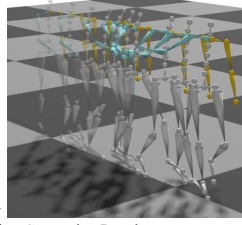 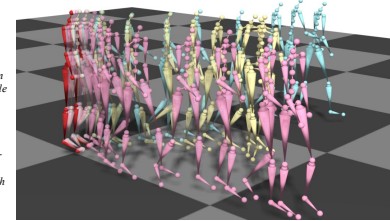

a) Text-to-Motion Generation Results      b) Motion Prediction Results

Figure 8: **More Analysis on EgoLM. a)** Qualitative results of text-to-motion generation. **b)** Qualitative results of motion prediction.

**Motion VQ-VAE.** Ablation studies on motion VQ-VAE are reported in Tab. 5. "PQ" denotes the number of codebooks. "CB" denotes the number of codebook entries. The first two lines indicate that significant improvements can be achieved simply by using product quantization. Additionally, increasing the number of codes and reducing code dimensions yields further enhancements.

**Larger Language Model.** We use GPT-2 Medium (345M) for most of our experiments to maintain efficiency. To further assess the potential of EgoLM in scaling up to larger LMs, we train with GPT-2 Large (1.5B) and report performance on TPV2T in Tab. 6. The improved scores indicate EgoLM is a scalable and versatile framework.

## 4.5 MORE APPLICATIONS

**Text-to-Motion Generation.** As part of our joint training, EgoLM is capable of generating motions from texts, as shown in Fig. 8 a). Even with lengthy prompts describing the upper and lower body separately, our model successfully generates motions that align with the inputs.

**Motion Prediction.** As a by-product of the motion pre-training, EgoLM can function as a motion predictor. As shown in Fig. 8 b), given motion prompts (the red skeleton in the left), subsequent motions can be randomly sampled. We show three different samples in different colors.

## 5 DISCUSSION

We propose EgoLM, an egocentric motion generalist model, that empowers egocentric motion understanding using LLMs. To address the challenge of limited wearer observation in egocentric perception, EgoLM integrates two complementary modalities to disambiguate the under-constrained scenarios. We also introduce multi-modal multi-task joint training to bridge gaps between different modalities and tasks, thereby implicitly connecting them and improving individual task performance. We hope our exploration of the fusion between egocentric perception and LLMs will inspire future research in contextual AI.

**Limitations.** Firstly, our motion tokenizer uses VQ-VAE, which introduces reconstruction errors and sets an upper bound for motion tracking performance. Additionally, during motion tracking training, the loss is calculated on discrete motion tokens rather than raw representations, which may further impact performance. Secondly, for motion narration, each egocentric video frame is compressed by the CLIP encoder into a one-dimensional vector, making it difficult for the model to accurately identify the objects the person is interacting with. Furthermore, as commonly observed in language models (Ji et al., 2023), EgoLM also experiences the hallucination problem.

**Potential Societal Impact.** While contextual AI presents opportunities for efficiency and societal advancement, the collection and analysis of human data may raise privacy concerns for users and those around them.

## REPRODUCIBILITY STATEMENT

We have thuroughly introduced our method in Sec. 3 as well as inference and experiment details in Appendix, which ensures the reproducibility. Moreover, the dataset used in this work is also publicly available at `https://www.projectaria.com/datasets/nymeria/`.

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

# APPENDIX

We provide more implementation details (Sec. A) and qualitative results (Sec. B) in this supplementary material. To better showcase our results, we also provide supplementary videos.

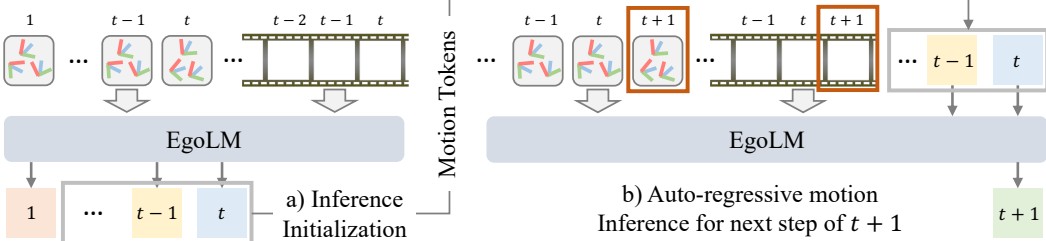

Figure 9: **Online Motion Tracking Inference.** For the new time step of $t+1$ with new data coming in, last motion tokens are combined with the new input tokens to decode the next motion token $t+1$.

## A   IMPLEMENTATION DETAILS

### A.1   AUTO-REGRESSIVE INFERENCE FOR MOTION TRACKING

At inference time, motion understanding is the same as the language model inference. For motion tracking, it usually requires online inference over a long period. With a language model, which is an auto-regressive model, it is straight-forward to perform online motion tracking. As shown in Fig. 9, firstly, an initialization over the first $t$ frames of data is required. When the new data frame $t+1$ comes in, the input conditions are updated accordingly. Then, it is not necessary to predict all the motion tokens from frame 2 to frame $t+1$. We take the previously generated motion tokens from frame 2 to frame $t$ as inputs and prompt the network to generate one more token for frame $t+1$.

### A.2   EVALUATION METRICS

For motion tracking, we use joint position errors and joint angle errors to evaluate the performance. Specifically, for the joint position errors, we first align ground truth skeletons and generated skeletons by the head positions only by translation. Then full body, upper body and lower body joint position errors are calculated separately. Joint angle errors are calculated on full body and root joints. For the evaluation of motion VQ-VAE in main paper Tab. 4, we apply widely adopted metrics for motion regression, *i.e.*, Mean Per-Joint Position Error (MPJPE) (Ionescu et al., 2013), Procrustes-Aligned (PA-)MPJPE (Kanazawa et al., 2018), and joint position acceleration (ACCL) error. For the motion understanding, we use standard NLP metrics, please kindly refer to corresponding papers for more details.

## B   MORE QUALITATIVE RESULTS

### B.1   THREE-POINTS MOTION TRACKING

We show four more visual examples of three-points motion tracking in Fig. 10, Fig. 11 and Fig. 12. AvatarPoser (Jiang et al., 2022) and BoDiffusion (Castillo et al., 2023) are solid baselines that perform well on easy walking cases, *e.g.*, upper example in Fig. 11. For the workout sequence, *i.e.*, lower example in Fig. 12, even only given three points of upper body, the distribution of lower body motion can be collapsed and generate reasonable motions that matches the ground truth. In Fig. 12, we demonstrate the effectiveness of including egocentric videos as inputs. Without any environment context, AvatarPoser and BoDiffusion often fail to distinguish standing and sitting down. We do not assume the knowledge of the head height over the floor, meaning that the three-points positions are normalized to the local coordinates of the first frame. Therefore, it is hard for baseline methods to disambiguate certain scenarios. We propose to introduce contexts using egocentric videos, which contains rich information about the environment and how the person is interacting with it. Therefore, our model can generate the most accurate motions by utilizing these information. For more visualization of three-points motion tracking, please kindly refer to our supplementary videos.

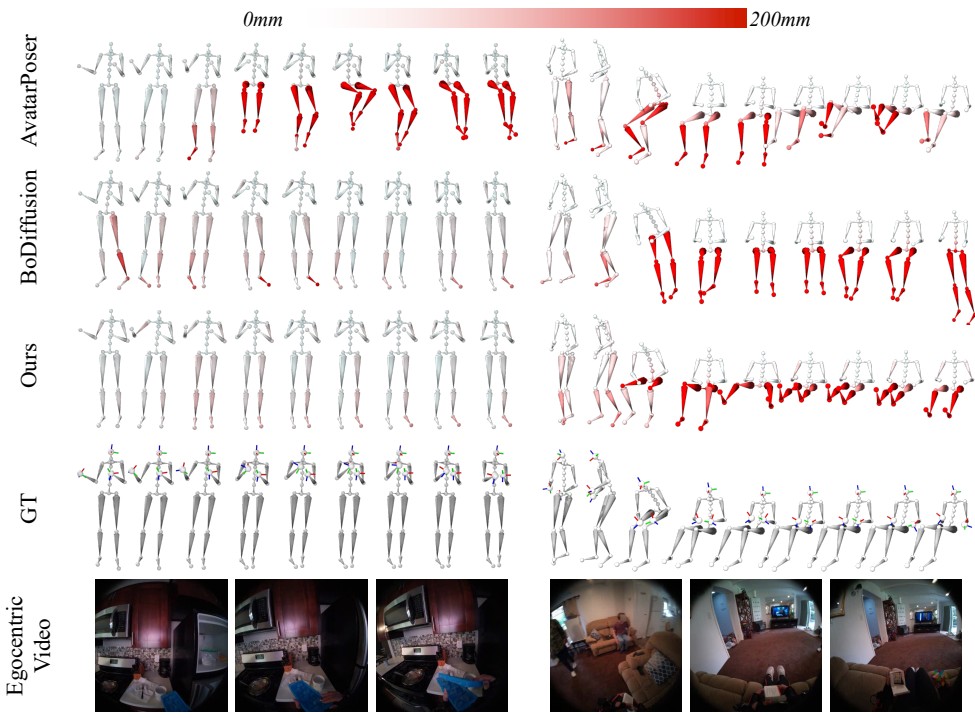

Figure 10: **Qualitative Results of Three-Points Motion Tracking.** Skeletons are color-coded by joint position errors.

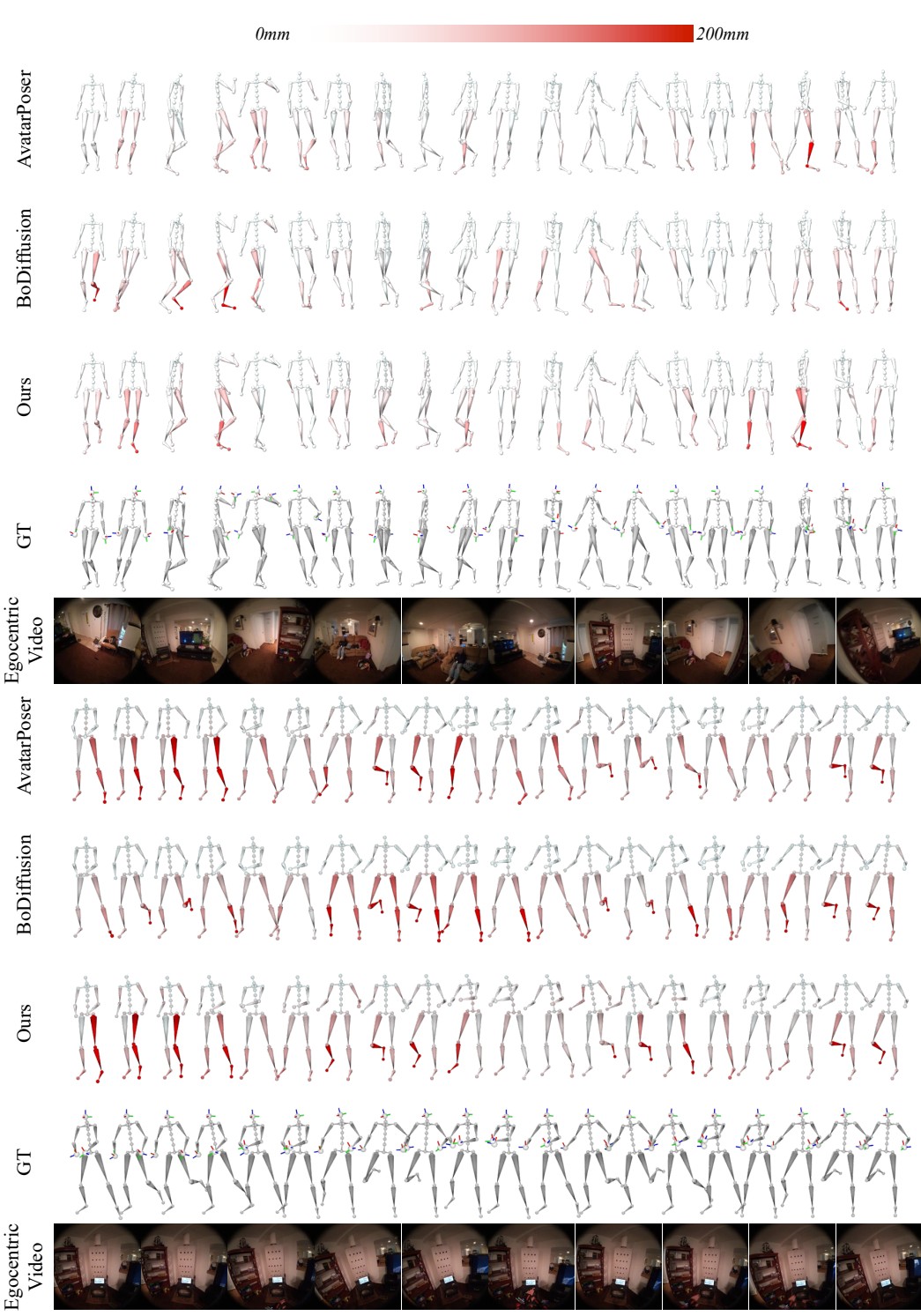

Figure 11: **Qualitative Results of Three-Points Motion Tracking.** Skeletons are color-coded by joint position errors.

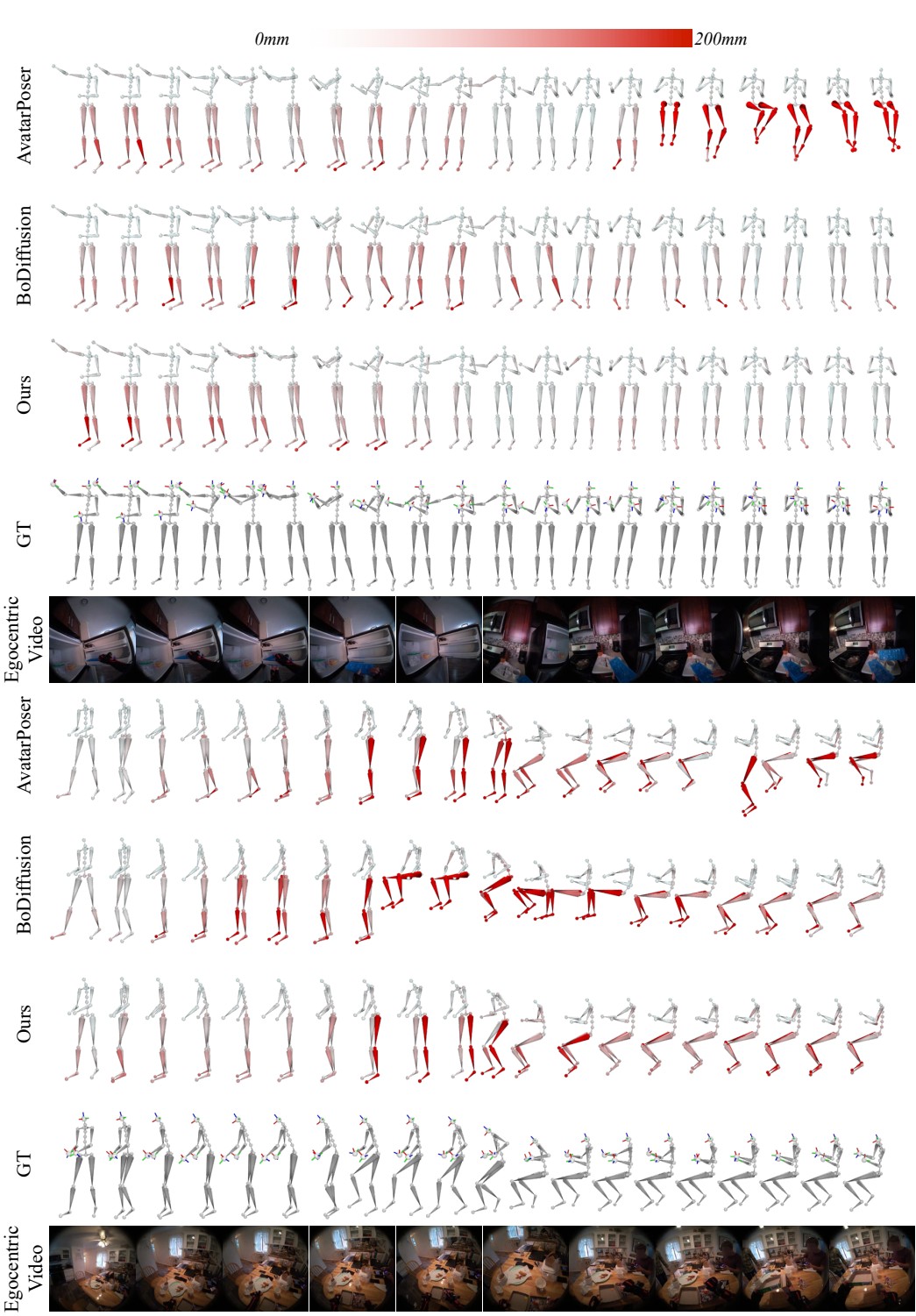

Figure 12: **Qualitative Results of Three-Points Motion Tracking.** Skeletons are color-coded by joint position errors.

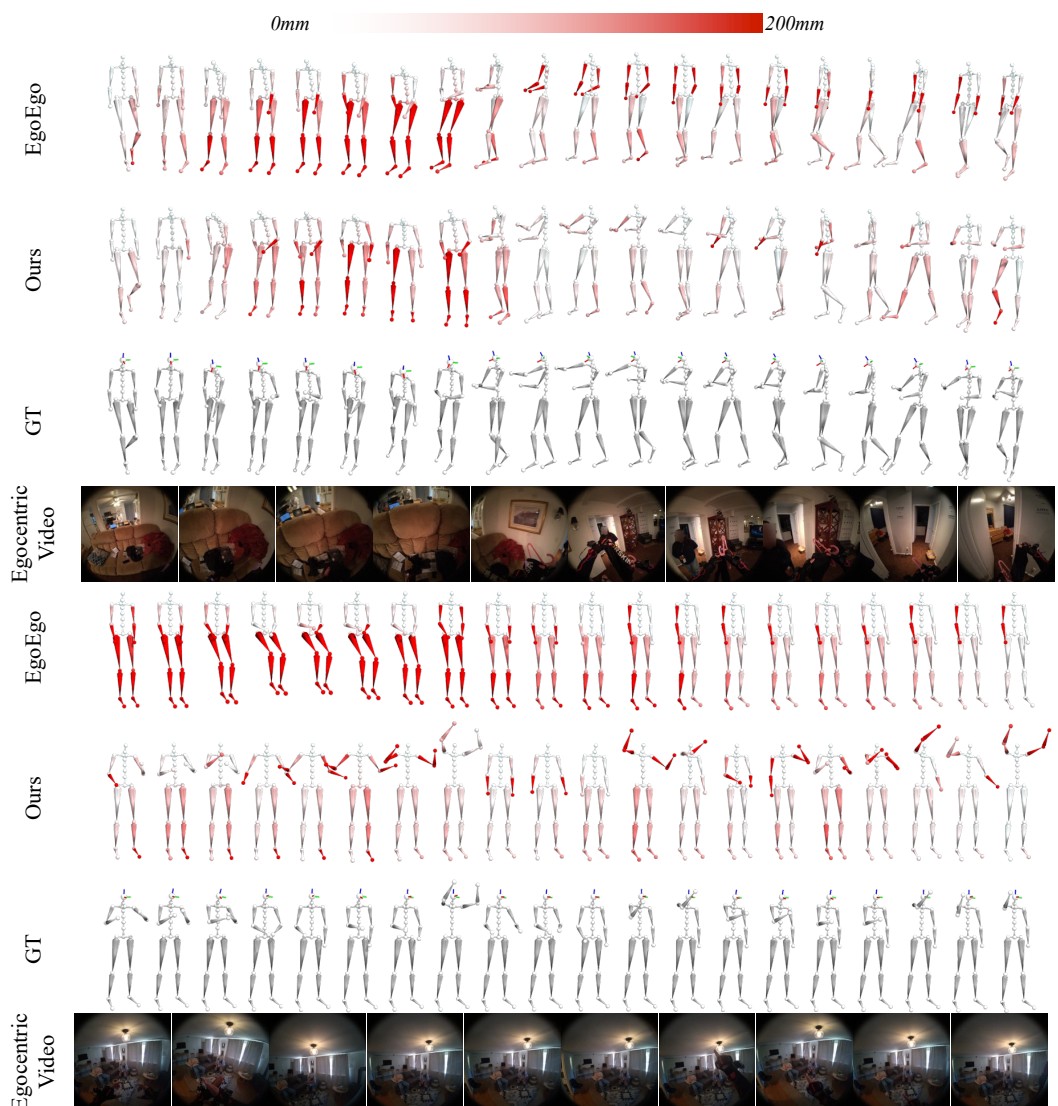

Figure 13: **Qualitative Results of One-Point Motion Tracking.** Skeletons are color-coded by joint position errors.

## B.2 ONE-POINT MOTION TRACKING

We show four more examples of one-point motion tracking in Fig. 13 and Fig. 14. The introduction of egocentric videos has two advantages. Firstly, similar to the case in three-points body tracking, the environment contexts in egocentric videos can disambiguate cases like standing and sitting. Secondly, specifically for one-point motion tracking, egocentric videos provide clues of hand positions. As shown in all four examples, when the person raises the arms in front of the body, hands would be visible in the egocentric videos, which helps the hand position tracking. Admittedly, high-level semantic information provided by CLIP (Radford et al., 2021) encoders cannot accurately track hand positions. Therefore, as shown in the lower example in Fig. 13, our method correctly generates arms moving in the air, but lacks accuracy. For more visual examples of one-point motion tracking, please kindly refer to our supplementary video.

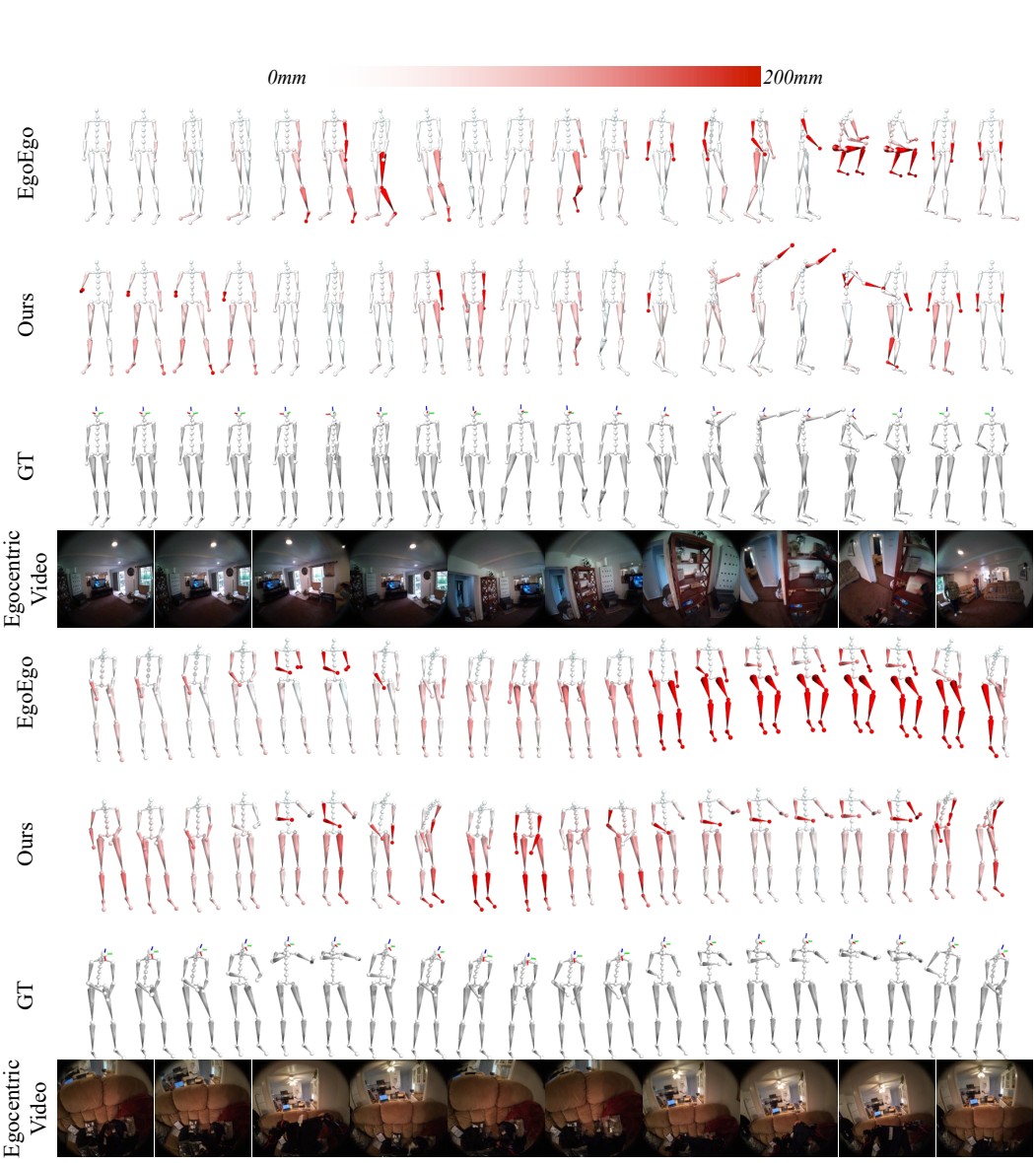

Figure 14: **Qualitative Results of One-Point Motion Tracking.** Skeletons are color-coded by joint position errors.

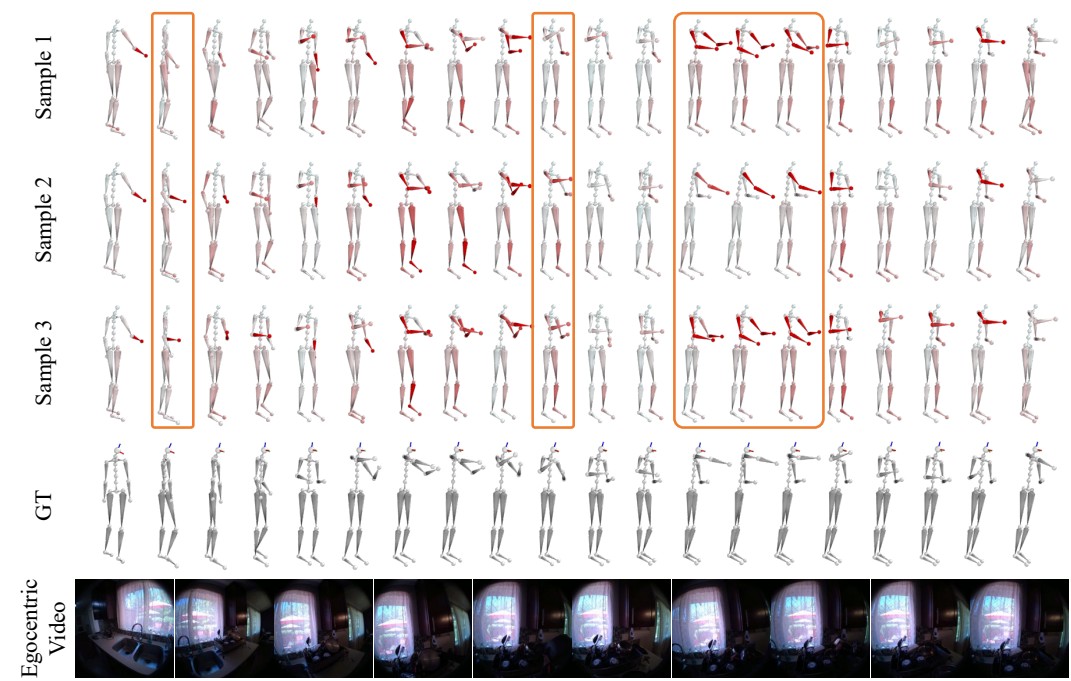

Figure 15: **Three Random Samples of One-Point Motion Tracking with Egocentric Videos as Inputs.** Since we use language models as our backbone, EgoLM has the ability to randomly sample outputs given the same inputs. Egocentric videos provide strong clues for hand positions, leading to less diversity as shown in the highlighted areas.

### B.2.1 MULTIPLE SAMPLES.

Note that EgoLM is essentially a generative model. Therefore, our model is capable of generating different samples with the same inputs. In Fig. 15, we show three random samplings on the same input one-point and egocentric video. When hands are not visible in the frame, *i.e.*, the left highlighted frame, hand positions are not constrained, and therefore shows high diversity across different samples. For the other highlighted frames, hands are visible in the egocentric videos, which helps to collapse the distribution of possible positions of hands. But as discussed above, our way of encoding egocentric videos cannot accurately track the hand positions. Therefore, our model also shows some diversity of hand positions in these cases.

To further demonstrate the diversity of our model, we also show three random samples from our one-point motion tracking model that does not take egocentric videos as inputs in Fig. 16. Lack of any indication of the hand positions, the upper body generation is even less constrained than that of the lower body and shows high diversity across three samples.

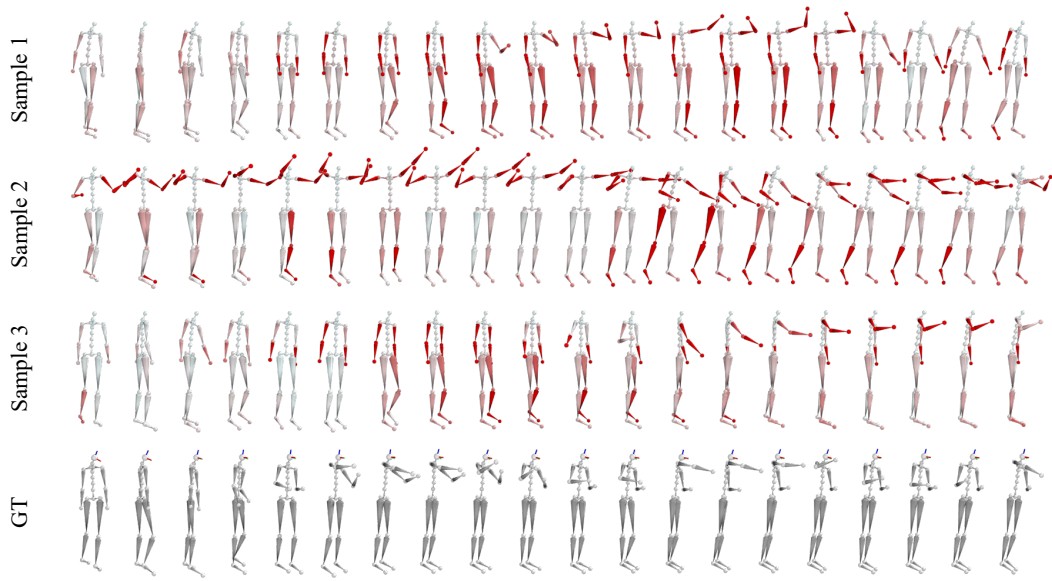

Figure 16: **Three Random Samples of One-Point Motion Tracking without Egocentric Videos as Inputs.** With only head poses as inputs, the generation of full body motion, especially upper body motions, is less constrained.

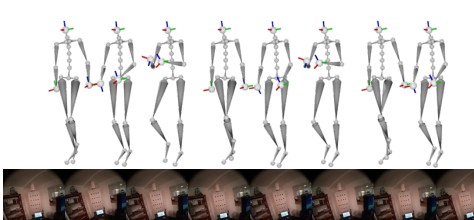

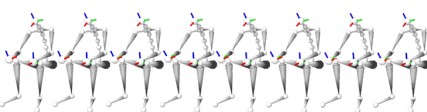

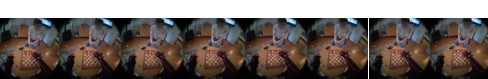

*TM2T*: *The person still* marches in place while facing his peers. *The person still* swings both of his hands up and down. *The person still marches in place with his left foot and right foot alternately. The person still repeatedly bends both of his legs alternately. The person still marches in place with his left foot and right foot alternately.*
*MotionGPT*: *The human swings his body to the right and swings back to the left while standing,* hunching his back *and* doing some exercise *in the living area with his colleagues. The human slightly* swings both of his arms back and forth on his side. *The human raises his right leg to his waist level then stretches and lowers it while his left foot is fixed on the floor.*
*V2T*: *The human is standing in the* living room *while* watching the television. *The person is* resting both arms on his sides. *The human* has both feet fixed on the floor.
*Ours*: *The person is swaying her body side to side while exercising in the* living area. *The person* repeatedly swings and bends both of her arms in front of her then lowers it down on her side. *The person* repeatedly raises both of her feet in front of her then lowers them down on the floor alternately.
*GT*: *The person is walking in place in front of the laptop. The human repeatedly bends both of her arm in front of her them lowers them down on her side. The human repeatedly steps both of her feet alternately.*

*TM2T*: *The person is* sitting on a chair *and* leaning backward *on the table while talking to her peers. The person is resting both of her arms on the table, lifts and bends her left arm as she* touches the table with her left hand. *The person is sitting with both legs bent and with both feet flat on the floor widely apart.*
*MotionGPT*: *The person is still* sitting on the chair with a hunched back *while* playing arcade and eating some chips. *The person's both arms are bent forward while holding and sliding the joystick with his left hand to the left then his right hand is on top of the buttons and clicks them with his right fingers. The person's both legs are still bent while sitting on the chair with both feet flat on the floor and slightly apart.*
*V2T*: *The human is* sitting *on the sofa and leaning forward while* arranging the chess pieces on the chessboard. *The person has both of her arms extended forward while picking up the chess pieces with her left hand and* puts down the chess piece with her right hand on the chess board. *The human is sitting with both feet fixed on the floor and shoulder-width apart.*
*Ours*: *The person is* sitting in front of the checkerboard. *The person is* extending his right arm toward the checkerboard while keeping his left arm on top of his leg. *The human is bending both of his knees while keeping both of his feet flat on the floor.*
*GT*: *The human is sitting in front of the table as he plays chess. The person is moving the knight with his right hand while his left hand remains resting on his leg. The human is bending both of his knees while keeping both of his feet flat on the floor.*

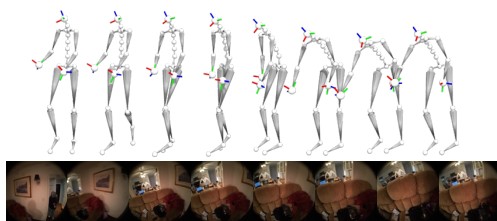

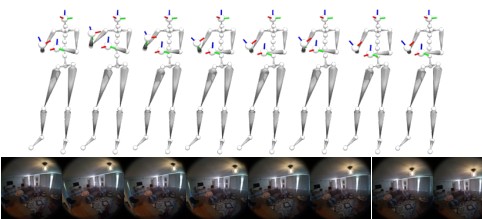

*TM2T*: *The person* walks towards the cabinet *then* bends forward to pick up and reach for the clothes. *The person* extends his right arm to pick up the clothes *from the* cabinet *then* bends his left arm to hold the clothes.
*MotionGPT*: *The person* bends forward *while standing in the living room. The person* extends her right arm to open the cabinet *and extends her left arm to grab the keys on the right. The person slightly bends both of her legs then steps her right foot forward while her left foot is fixed on the floor.*
*V2T*: *The human* walks towards the couch *and* bends down *while putting down the piece of clothing. The person extends both of her arms to* pick up and put down *the piece of* clothing with her right hand *while* holding the clothes with her left hand. *The human steps both of her feet forward alternately.*
*Ours*: *The human* walks towards the sofa *then* slightly leans forward *to put down the folded piece of clothing. The person* extends her right arm *to* put down *the folded piece of clothing on the sofa, then* extends her left arm to pick up *another piece of clothing on the sofa. The human is stepping both of her feet forward alternately then bends both of her legs to support her body.*
*GT*: *The person bends his body to get another clothes on the sofa. The person extends his right arm to get the clothes with his right hand then raises his left arm to hold the clothes with his left hand. The person steps both feet forward towards the sofa.*

*TM2T*: *The person is still* standing straight in front of the table *while playing the board game with his peer. The person's* both arms are still bent *forward while both hands are still* holding the edge of the knife.
*MotionGPT*: *The human still* stands near the closet. *the human still* holds the hanger *with his left hand and his right hand holding the hanger. The person still stands with his feet slightly apart.*
*V2T*: *The person is* standing straight in the living area *with his colleagues while doing some exercise. The person raises both of his arms straight* above his head *from the back then lowers them in front and* rests them on his side. *The person is standing with both feet apart and fixed on the floor.*
*Ours*: *The person is* standing in the living area. *The human repeatedly* swings both of his arms in front of him *and in front of his stomach. The person is standing with both feet fixed on the floor.*
*GT*: *The human is standing still in front of his colleagues in the living room while playing charades. The person is slightly raising and lowering both of his arms to gesticulate. The human is resting both of his legs fixed to the floor.*

Figure 17: **Qualitative Results of Motion Narration.** We use green to highlight correct parts in the answers while red for wrong ones.

## B.3 MOTION NARRATION

We show eight more examples of motion narration in Fig. 17 and Fig. 18. Similar to the main paper, we use green to highlight correct parts in the answers and red for wrong answers. Similar to the observation made in the main paper, even though TM2T (Guo et al., 2022b) and MotionGPT (Jiang et al., 2024) have access to the full body motion, the generated narrations are reasonable but completely wrong if consider the environment context. For example, in the upper right example in Fig. 18, given the simple walking sequence, both TM2T and MotionGPT can correctly understanding that the person is walking forward. But they all give the wrong answers about the places the person is walking in. Thanks to the egocentric videos, our model successfully produces the correct description as "walking towards the beds".

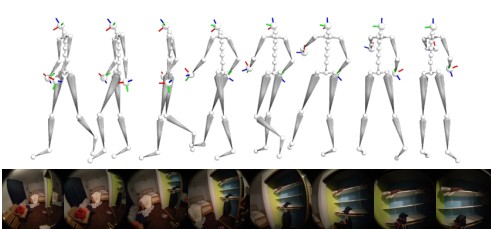

*TM2T*: *The person walks out of the bedroom then turns to the left to enter another bedroom. The person rests both her arms on her sides. the person is stepping forward with her right and left legs alternately. The person is stepping forward with her right and left legs alternately. The person then steps forward with her right feet.*
*MotionGPT*: *The person walks in the garage. the person sways his hands on the sides. The human extends both legs forward alternately.*
*V2T*: *The human walks towards the bedroom. The human slightly sways her hands on her sides. The human takes four steps towards the bedroom.*
*Ours*: *The human walks towards the door. The human puts down her right arm and sways both hands on the side. The person extends both legs forward alternately.*
*GT*: *The person walks towards the door. The person walks towards the door. The person rests his left arm on the side and he raises his right arm while holding the hanger with his right hand. The human extends both legs forward alternately.*

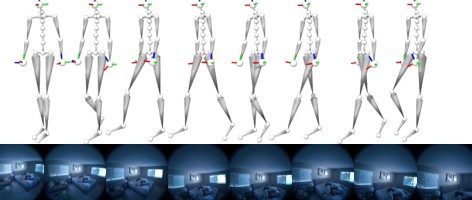

*TM2T*: *The person is walking forward in the pathway then she slightly leans forward as she sits on the pathway. The person alternately swings both hands on her sides while both arms hang naturally at her sides.*
*MotionGPT*: *The human is walking forward while looking at the office surrounding. The human has her both arms swaying them back and forth. The human extends both legs forward alternately.*
*V2T*: *The person is walking forward towards the bed. the person rests both arms on her sides. The person is extending both her legs forward alternately.*
*Ours*: *The human is walking towards the bed. The person is resting both of her arms beside her. The person is extending both of her legs forward alternately.*
*GT*: *The person walks towards the bed. The person slightly swings both of her arms back and forth. The person steps both of her legs forward alternately.*

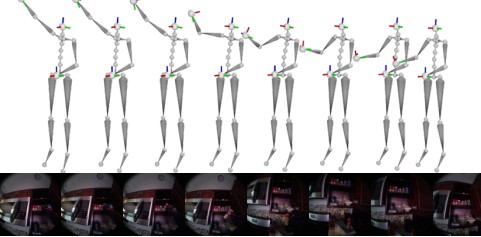

*TM2T*: *The person walks towards the door then leans forward as he tucks in the chair and stands in front of the door to open it. The person's right arm is swinging back and forth on his side while his left arm is bent and his left hand holding the top railings then pushes the door open with his left hand.*
*MotionGPT*: *The person is walking forward towards the shower room, pauses on the shower room and then leans forward to put down the towel on the shower curtain holder. The person is bending both of his arms and then extends his left arm forward to put down the towel on the shower curtain holder. The person is alternately stepping both of his feet forward.*
*V2T*: *The person straightens up as she slightly turns to the left while walking towards the closet. The person keeps holding the clothes with her bent left arm as she lowers down and slightly raises her right arm and then she bends it back. The person steps both of her feet forward alternately.*
*Ours*: *The human turns clockwise as she walks towards the closet to put the clothes on the top shelf in the bedroom. The human is holding the clothes hanger with both of her bent arms in front of her then she extends her left arm froward and grabs the clothes hanger with her left hand. The human turns her right foot to the right, steps her left foot forward then slightly moves her right foot forward.*
*GT*: *The human walks towards the closet. The human raises his left arm to grab the clothes while he holds the hanger with his right hand. The person extends both legs forward alternately.*

*TM2T*: *The person is standing in front of the door. the person is raising his left arm and is resting his right arm on his side. The person bends both of his legs while resting on the floor.*
*MotionGPT*: *The person stands in the bedroom while talking to her colleague. The human is resting and bending her left arm in front while she lowers down her right hand before touching the wall with her right hand. The person stands with both feet fixed on the floor.*
*V2T*: *The human is standing straight while picking a condiment jar in the hanging cabinet. The human grabs a condiment jar with her right hand and flips up the other condiment jar in front of her with right hand and then she bends and slightly lowers down her right arm. The person is standing with both feet fixed on the ground.*
*Ours*: *The person is standing in front of the hanging cabinet and slightly leaning forward while picking up a condiment jar. The person is extending her right hand forward, picks up the condiment jar cover then puts it down again on the top of the hanging cabinet while resting her left arm on her side. The human is standing with both of her legs parallel to each other and both of her feet spread slightly apart.*
*GT*: *The person is standing on tiptoes while checking inside the cupboard. The human grabs and places the bottle down on the countertop with her right hand while her left hand is resting on the countertop. The human is standing on tiptoes with both feet as she reaches inside the cupboard.*

Figure 18: **Qualitative Results of Motion Narration.** We use green to highlight correct parts in the answers while red for wrong ones.

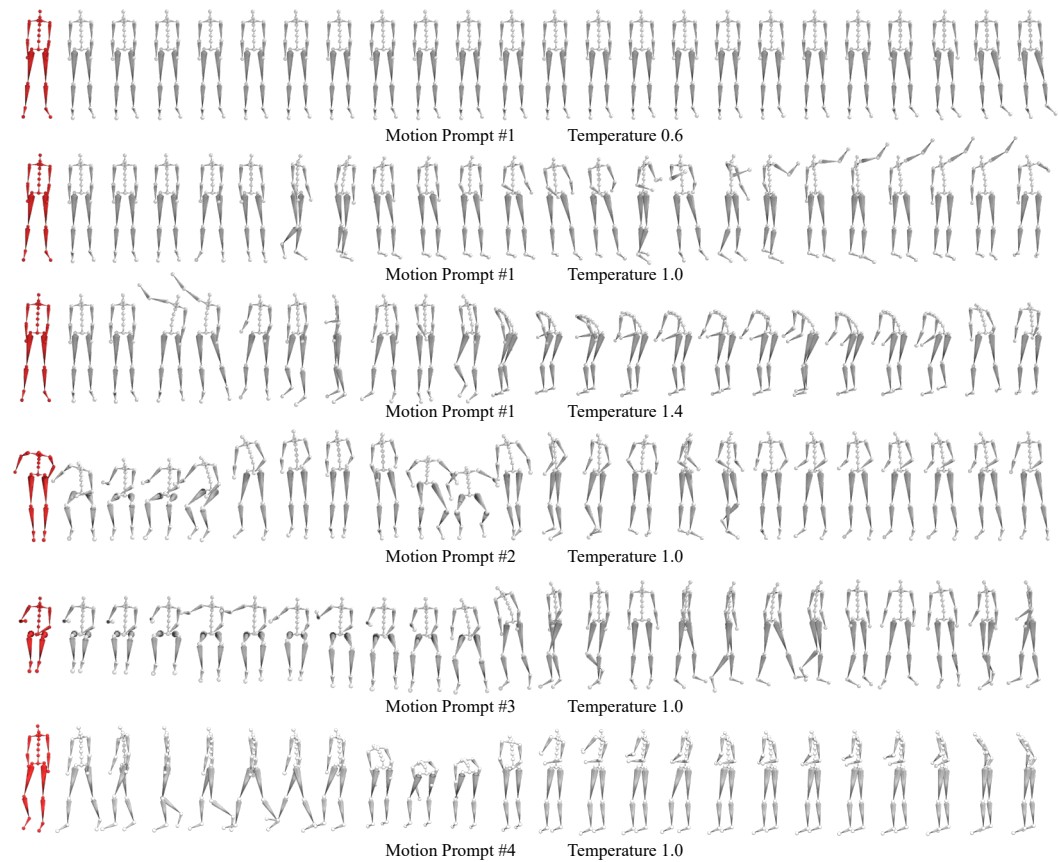

Figure 19: **Qualitative Results of Motion Prediction.** The first skeletons in red are input motion prompts. The following motions are randomly sampled auto-regressively from our motion pre-training network.

## B.4 MOTION PREDICTION

As a by-product of the second stage of our training pipeline, motion pre-training, we build a motion prediction network. Given leading motions as the prompts, our model is capable of auto-regressively sample motions that complete the motion prompts. As shown in Fig. 19, the first three samples show three different samples given the same motion prompt. We can increase the intensity of the generated motions by increasing the temperature. The last three samples show three random samples given various motion prompts, *e.g.*, bending forward, sitting down and standing.

