# OpenReview forum: "EgoLM: Multi-Modal Language Model of Egocentric Motions"
_ICLR.cc/2025/Conference — ICLR 2025 Conference Withdrawn Submission_

### Official Review · Reviewer_4ygP · 2024-10-31

**Soundness:** 2
**Presentation:** 3
**Contribution:** 2
**Rating:** 5
**Confidence:** 4

**Summary:**

This paper proposed a new work on egocentric video understanding and human motion recovery with sparse headset pose data. By combining the language and motion modalities, the authors built a new pipeline to train the LLM with motion data into a language-motion LM. On motion recovery, egocentric video captioning, etc., the proposed method achieves decent performances compared to baselines.

**Strengths:**

+ The data focused on by this paper means a lot in the future world, as the VR equipment will be an important entry of human life data.

+ Reconstructing human motion from the sparse 6D pose data is non-trivial.

**Weaknesses:**

- The method contribution: most of the method designs in this work are common practice from the other VLM or MLLM works, like instruction tuning tricks, VQ-VAE encoding, and training, joint space training, multi-modal tokenization, etc. Please add a comparison between previous works, or which part is your unique contribution.

- Though this work focuses on the sparse headset pose data, many existing 3D pose datasets on videos (like BABEL) can also be formalized as similar benchmarks. At least, these data can be considered as the data supplementary in the pre-training. Moreover, there are also many baselines from the other tracks that can be adopted here, like different recovering methods from incomplete inputs.

- Even the motion captioning part, many datasets like BABEL can also be seen as a testbed to verify the proposed method.

Overall, both the method and experiment parts have a lot of room for improvement.

**Questions:**

1. If using more 3d motion data from related datasets, e.g., transferring them into the data used in this work, can the performance boost?

2. The joint training affect on LM? Why not use more recent open-sourced LLM like Llama?

3. The effects of different headsets/sensors?

4. Tab 2: seems that ours w/o video performs not well. Could the authors provide more insight?

---

### Official Review · Reviewer_Qotq · 2024-11-03

**Soundness:** 2
**Presentation:** 3
**Contribution:** 2
**Rating:** 6
**Confidence:** 3

**Summary:**

EgoLM is a framework designed to make sense of motion data from wearable devices by combining video and sensor inputs.
It tackles a range of tasks, from generating motion narratives and text descriptions to synthesizing motion from sparse data, all while using rich contextual cues.
The authors demonstrate EgoLM’s versatility and effectiveness in handling complex motion understanding challenges, showing its potential as a powerful tool for HMD.

**Strengths:**

EgoLM stands out for its versatility, unifying diverse egocentric motion tasks such as narration, generation, and text synthesis in a single framework.
The model effectively combines video and sensor inputs, capturing rich contextual information that enhances motion understanding.
Extensive experiments demonstrate EgoLM’s adaptability for wearable technology applications.

**Weaknesses:**

While I believe that using an egocentric body model with a language model is a promising research direction, I have concerns about its practicality in real-world applications due to potentially slower inference speeds. If inference time is slow, it may hinder real-time motion tracking capabilities, even with an online setup.

Regarding motion tracking, state-of-the-art methods such as [1], [2], and [3] were available at the time of submission to ICLR. Comparing EgoLM's results to these more recent methods could provide a more comprehensive evaluation. Is there a specific reason for selecting BoDiffusion and AvatarPoser for comparison instead?

Given EgoLM's significant performance boost when using image features, I also suggest comparing inference times with and without image features to see if the input type affects performance. Does inference time remain consistent regardless of input type?

[1] Du, Y., Kips, R., Pumarola, A., Starke, S., Thabet, A., & Sanakoyeu, A. (2023). Avatars grow legs: Generating smooth human motion from sparse tracking inputs with diffusion model. In Proceedings of the IEEE/CVF Conference on Computer Vision and Pattern Recognition (pp. 481-490).

[2] Feng, H., Ma, W., Gao, Q., Zheng, X., Xue, N., & Xu, H. (2024). Stratified Avatar Generation from Sparse Observations. In Proceedings of the IEEE/CVF Conference on Computer Vision and Pattern Recognition (pp. 153-163).

[3] Zheng, X., Su, Z., Wen, C., Xue, Z., & Jin, X. (2023). Realistic full-body tracking from sparse observations via joint-level modeling. In Proceedings of the IEEE/CVF International Conference on Computer Vision (pp. 14678-14688).

**Questions:**

Questions Regarding Motion Tracking
- Could you clarify the statement, "All methods are evaluated using batch inference, where every 60 frames are processed independently"? I initially assumed that inference was performed in an online manner with a 1-frame sliding window, so this sentence is a bit confusing to me.
- Is the configuration setup for AvatarPoser and BoDiffusion the same as EgoLM in Table 2, 1 frame sliding winodw?
- I believe that velocity error would be an informative metric for ego-body tracking. Could you share the velocity error results from Table 2 for both the baselines and EgoLM?

Code Availability
- Will the code and model be made publicly available?

---

### Official Review · Reviewer_YFJL · 2024-11-03

**Soundness:** 3
**Presentation:** 3
**Contribution:** 3
**Rating:** 6
**Confidence:** 3

**Summary:**

The paper focuses on improving egocentric motion understanding using large language models. When using smart wearable devices, egocentric information about the user's environment and interactions can be collected, however it lacks information about the wearer's body pose and kinematic motion of the various body parts. The paper's insight is that the egocentric videos can help to disambiguate the lower body motion and sparse motion sensors can give information when the egocentric video is identical.

In the method, the sparse motion sensor data and visual information are aligned with the language representations of the LLM to give a generated motion and a motion narration text as output. This method is trained as a multi-modal multi-task manner to understand motion on a semantic and kinematic levels. The text tokens of the prompt, clip embeddings, and three points of the motion are concatenated together and passed to the language model to generate the output.

The method is compared on the motion tracking and motion narration tasks with the AvatarPoser and Bodiffusion baselines and show a significant improvement in terms of the tracking metrics (full, lower, upper,  joint position, joint angle) and narration metrics (bleu score, rouge).

**Strengths:**

1. The paper shows exhaustive evaluation with the unimodal baselines and compares both the multi-modal and uni-modal versions of the method.
2. The paper leverages multiple modalities of sensor data and visual data for better motion tracking.
3. The qualitative results are exhasutive.

**Weaknesses:**

1. How does the method deal with scenarios when both the video and sensor data become ambiguous? For example, if the person is using the treadmill and faces the white wall, can the sensor data confuse between walking and running?
2. How is the correspondence/association established between the two modalities - clip embedding and pose data?

**Questions:**

1. Is there any noise in the sensor data?

---

### Official Review · Reviewer_8i1z · 2024-11-03

**Soundness:** 3
**Presentation:** 3
**Contribution:** 2
**Rating:** 3
**Confidence:** 4

**Summary:**

This work introduces EgoLM, a framework for egocentric motion understanding using a language model that processes multi-modal data inputs. To improve egocentric motion comprehension, the authors propose a method to align motion and language distributions. They also introduce interaction-tuning-based multi-modal training. Using EgoLM, the paper addresses two tasks: motion tracking and motion narration. Through experiments, the paper demonstrates the effectiveness of the proposed approach in egocentric motion-related tasks.

**Strengths:**

- This paper presents a new framework named EgoLM for understanding egocentric motion.
- The figures are high-quality and aid in understanding the proposed approach.

**Weaknesses:**

- Limited technical novelty: Sections 3.2 and 3.3 are almost identical to T2M-GPT, as they discuss training motion VQ-VAE and training an autoregressive LM on motion tokens. This makes it challenging to consider these sections as novel contributions. Section 3.4, which introduces multi-modal multi-task training, is the only unique technical contribution; however, it merely combines different modalities to instruction-tune the LM without introducing new technices.
- The proposed framework does not include design for egocentric video. It lacks a specific algorithm tailored to egocentric views and instead emphasizes combining different modalities, which does not need to be egocentric applications.
- The model is only compared to baselines that do not include video. From the experiments, it seems that the performance improvement may be attributed to the addition of a video rather than the proposed framework itself, as the model’s performance is almost similar to the baselines without video. Experiments under identical conditions are needed to clearly establish the framework’s contribution.

**Questions:**

- In Table 2, EgoEgo can also take ego video as an input. Is there a reason why you didn’t compare the performance of EgoEgo 1pt + Video?

---

### Note · Authors · 2024-11-14

**Comment:**

We thank all reviewers for their efforts and thoughtful comments.

**Withdrawal Confirmation:**

I have read and agree with the venue's withdrawal policy on behalf of myself and my co-authors.